



# Ground-based vertical profile observations of atmospheric composition on the Tibetan Plateau (2017-2019)

Chengzhi Xing[1], Cheng Liu[2,1,3,5,6,*], Hongyu Wu[1], Jinan Lin[1], Shuntian Wang[1], Meng Gao[4,7,*]

[1]Key Lab of Environmental Optics & Technology, Anhui Institute of Optics and Fine Mechanics, Hefei Institutes of Physical Science, Chinese Academy of Sciences, Hefei, 230031, China

[2]Department of Precision Machinery and Precision Instrumentation, University of Science and Technology of China, Hefei, 230026, China

[3]Center for Excellence in Regional Atmospheric Environment, Institute of Urban Environment, Chinese Academy of Sciences, Xiamen, 361021, China

[4]Department of Geography, State Key Laboratory of Environmental and Biological Analysis, Hong Kong Baptist University, Hong Kong SAR, China

[5]Key Laboratory of Precision Scientific Instrumentation of Anhui Higher Education Institutes, University of Science and Technology of China, Hefei, 230026, China

[6]Anhui Province Key Laboratory of Polar Environment and Global Change, University of Science and Technology of China, Hefei, 230026, China

[7]Hong Kong Branch of Southern Marine Science and Engineering Guangdong Laboratory (Guangzhou), Hong Kong, China

*Correspondence to:* Cheng Liu (chliu81@ustc.edu.cn) and Meng Gao (mmgao2@hkbu.edu.hk)

**Abstract.**

The Tibet Plateau (TP) plays an essential role in modulating regional and global climate, and its influence on climate is affected also by human-related processes, including changes in atmospheric composition. However, observations of atmospheric composition, especially vertical profile observations, remain sparse and rare on the TP, due to extremely high altitude, topographical heterogeneity and grinding environment. Accordingly, the forcing and feedback of atmospheric composition from rapidly changing surrounding regions to regional environmental and climate change in the TP remains poorly understood. This paper introduces a high time-resolution (~15 min) vertical profile observational dataset of atmospheric composition (aerosol, NO2, HCHO and HONO) on the TP for more than one year (2017-2019) using a passive remote sensing technique. The diurnal pattern, vertical distribution and seasonal variations of these pollutants are documented here in detail. The sharing of this dataset would benefit scientific community in exploring source-receptor relationships, forcing and feedback of atmospheric composition on the TP to regional and global climate. It also provides potentials to improve satellite retrievals, and to facilitate the development and improvement of models in cold regions. The dataset is freely available at Zenodo (http://doi.org/10.5281/zenodo.4911384; Xing et al., 2021).



## 1 Introduction

The Tibetan Plateau (TP) and the surrounding Hindu Kush Himalayan mountains are referred to as the world's "Third Pole" due to their vast wealth of areas and glaciers. It is the highest and largest plateau in the world with a ~4000 m averaged elevation. Headwaters of six major rivers in Asia start on the TP, (Kang et al., 2010; Cheng and Jin, 2013), and it thus acts the role of "Water Tower of Asia", determining the survival and development of more than 2 billion people (Xu et al., 2008; Immerzeel et al., 2010; Gao et al., 2019; Kang et al., 2019). A number of studies have suggested that the TP is an important driving force to thermally and dynamically affect global and regional climate (Yanai et al., 1992; Liu et al., 2007; Boos et al., 2010). In contrast to the surrounding cold middle troposphere, the TP receives strong solar radiation at the surface owing to its topographic characteristics, leading to its critical role in the evolution and variability of South and East Asian monsoons (Wu et al., 2007, 2012). It also acts as a natural barrier that inhibits northward moisture transport (Dong et al., 2017) and splits subtropical westerlies (Bolin et al., 1950). The TP is usually regarded as one of the most sensitive areas responding to global environmental and climate change (Peng et al., 2012), and can amplify the changes to global scale (Kang et al., 2010).

Recent decades have witnessed increased warming, together with retreat of glaciers and early melting of snow, in the TP (He et al., 2003; Lau et al., 2010; Xu et al., 2016), leading to serious side issues, including soil erosion, floods, etc. (Shrestha et al., 1999). As the warming rate of the TP is much faster than the greenhouse warming rate, it was argued that other factors besides greenhouse warming might have resulted in accelerated warming over the TP (Kang et al., 2000; Ramanathan et al., 2007). The TP was usually considered as a background region of atmospheric composition due to low population density and less intense human activities. However, studies in the past decade suggested that transport of pollutants from south Asia, east Asia and southeast Asia cannot be ignored (Cong et al., 2009; Yao et al., 2012; Zhang et al., 2015; Kang et al., 2019). Ample evidence has demonstrated the significance of atmospheric composition in influencing the retreat of glaciers on the TP (Xu et al., 2009), while large uncertainties remain in understanding the sources, mainly owing to lack of long-term observations that obstructed by complex topography and harsh environment (Cong et al., 2015; Barnett et al., 2005; Pu et al., 2007; Bolch et al., 2012; Kang et al., 2016).

To fill these gaps, a series of field campaigns were carried out in recent years to unveil the features of weather and atmospheric composition over the TP, especially after the "second comprehensive scientific expedition to the Qinghai-Tibet Plateau" plan (Wang et al., 2021; Che et al., 2021; Liu et al., 2021). The Institute of Tibetan Plateau Research, Chinese Academy of Sciences (ITP, CAS) established six long-term stations to measure both meteorological and micrometeorological variables over the TP (Ma et al., 2020). China National Environmental Monitoring Center (CNEMC) has also established an atmospheric composition monitoring network (Gao et al., 2020b; Liang et al., 2017) on the TP to continuously monitor the ground-level concentrations of $PM_{2.5}$, $PM_{10}$, $NO_2$, $SO_2$, $O_3$ and CO since 2013, covering Lhasa, Shigatse, Qamdo, Nyingchi, Shannan, Nagqu and Ngari. Satellite retrievals (i.e. OMI, MODIS and CALIPSO) have also been widely applied to elucidate the spatial and temporal variation of atmospheric processes over the TP (Zhu et al., 2019; Li et al., 2020). Although the vertical profiles of aerosols were revealed with CALIPSO (Huang et al., 2007), the relatively long



revisiting (~ 16 days) hinders understanding of processes at smaller scales. Accordingly, the current atmospheric composition monitoring platforms and freely available datasets are inadequate to fully understand the sources and impacts of atmospheric composition on the TP, as the formation, aging and transport processes also occurs above the ground (Hindman et al., 2002; Duo et al., 2018; Huang et al., 2007). If we only use satellite column or ground-level observations to infer

transport fluxes of atmospheric trace gases, the relative significance of local and transport contributions could be overvalued/downplayed (Hu et al., 2020; Liu et al., 2021a). In addition, the low air density, land surface heterogeneity, relatively cold temperature and strong solar radiation on the TP cause relatively higher planetary boundary layer (PBL) compared to its surrounding lowlands (Yang et al., 2003; Seidel et al., 2010). It promotes the exchange of atmosphere between PBL and stratosphere, which has important impacts on atmospheric chemistry (Skerlak et al., 2014). The lack of

sufficient vertical profile observation data limits the understanding of atmospheric composition on the TP and its impact on the global climate.

Limited field experiments were conducted using balloon and lidar on the TP in recent years (Wu et al., 2016; Dai et al., 2018; Fang et al., 2019; Zhang et al., 2020). The instruments used are costly and very limited atmospheric composition species (mostly $O_3$) can be measured. On the other hand, these observations are not open for sharing or very limited available

upon request (i.e. only data during specified periods are provided). The coarse temporal resolution of these data is not sufficient for evaluations of chemical transport modeling and climate modeling (Gao et al., 2020a). Multi-axis differential optical absorption spectroscopy (MAX-DOAS) uses scattered sunlight as the signal source, enabling it to achieve low-cost and continuous measurement of vertical profiles of atmospheric composition.

Here we describe and provide access to a high time-resolution dataset of vertical profiles of atmospheric composition over

the TP for more than one year. This database will play a critical role in improving satellite retrievals and numerical modeling of atmospheric composition over the TP. The uncertainties of air mass factor (AMF) caused by the high surface albedo and the absence of a priori profile on the TP could lead to large uncertainties in satellite retrievals of trace gases. Previously, we have demonstrated that using AMF calculated with MAX-DOAS measured $NO_2$ vertical profiles could remarkably improve the accuracy of retrievals of $NO_2$ column densities (VCD) (Liu et al., 2016; Xing et al., 2017). On top of that, due to complex

terrain and weather conditions, uncertainties in emission inventories, and imperfect model parameterization, chemical transport models are difficult to capture the vertical structures of atmospheric composition on the TP (Yang et al., 2018b), the uncertainties of which could be constrained through assimilation of observations from this dataset. Sect. 2 describes the observation site, MAX-DOAS instrument and retrieval algorithms. The vertical profiles of aerosol, $NO_2$, HCHO and HONO, and their diurnal and monthly variations are introduced in Sect. 3. Sect. 4 and Sect. 5 present the availability of this dataset

and a summary, respectively.



## 2 Experimental setup

### 2.1 Description of the monitoring site

The MAX-DOAS instrument was setup at the Qomolangma Atmospheric and Environmental Observation and Research Station, CAS (QOMS) (4276 m, 28.21ºN, 86.56ºE), which is located at the bottom of the lower Rongbuk valley (Fig. 1). The geomorphic of this station is mainly alpine covered by sandy soil, and sparse vegetation exsits. During the summer monsoon season, abundant moisture is prevailed to the CAS (QOMS) station from the Indian Ocean, increasing humidity and precipitation there. The maximum humidity was observed from July to August, and lower values usually appeared from

December to February (Zhang et al., 2018; Chen et al., 2018). To the south of the measurement site is the South Asian subcontinent, covering Nepal, India and Bangladesh. The South Asian subcontinent is currently one of the most polluted regions in the world. Averaged spatial distributions of satellite monitored aerosol optical depth (AOD), $NO_2$ and HCHO from December 2017 to March 2019 are displayed in Fig. S1. Elevated AOD, $NO_2$ and HCHO values are found in the southern foothills of the Himalayas compared to those in the Tibetan plateau. Local circulations, such as valley winds,

closely connect the near-surface atmosphere with free atmosphere on the north side of Mt. Everest, and South Asia summer monsoon could drive pollutants from the South Asian subcontinent to influence atmospheric composition over the TP. Accordingly, the CAS(QOMS) site acts as a terrific location to monitor the vertical distribution of atmospheric composition, and to explore the sources and impacts of these pollutants.

### 2.2 Description of the MAX-DOAS instrument

The MAX-DOAS instrument at the CAS (QOMS) station was operated from December 2017 to March 2019. It consisted of two key parts, namely the telescope scanner unit and the spectrometer unit (Fig. 2). Temperature stabilization of both spectrometers, USB communication and control-electronics for additional sensors to record temperature, pressure, inclination, and the telescope elevation stepper-motor were also incorporated.

The telescope unit contained motor, electronics, MEMS inclination sensor and optical components. A quartz-glass tube was

used to protect the rotating prism, which reflected the light through a lens onto the fibre at the end of an adjustable tube. This unit was controlled by a stepping motor to collect the scattered sunlight at different elevation angles, which was then transmitted via a glass fiber bundle to the spectrometer unit. The stepping motor had a precision of 0.01º, and the field of view (FOV) of telescope was less than 0.3º. Telescope scanner unit covered elevation angles from -10º to 180º, with angle accuracy < 0.1º. Moreover, an azimuth motor covering 0º-180º was installed in the telescope scanner unit to observe

atmospheric composition at different azimuth angles. In this study, we set the elevation angle sequence to 1, 2, 3, 4, 5, 6, 8, 10, 15, 30 and 90º, and exposure time of each individual spectrum to 1 min. The telescope unit was pointed to an azimuth angle of 53º.



The spectrometer unit included two AvaSpec-ULS2048L spectrometers (UV: 300-460 nm, visible: 460-630 nm) with a 0.6

nm spectral resolution. A cooling unit was used to control the spectrometer temperature. Typical instrumental stray light was

$< 0.05\%$, root mean square (RMS) of $1\times10^{-4}$ (visible) and $2\times10^{-4}$ (UV) for $\approx$1000 scans around noon. A Peltier element

was mounted on the spectrometer housing to cool and heat the spectrometer, and the temperature was stabilized at 20℃

with low fluctuations of $< 0.05$ ℃. We also equipped a charge-coupled device camera (Sony ILX511 with 2048 pixels) to

the spectrometer to convert analog signal to digital signal. Moreover, the dark current and electric offset were recorded at

night to correct the observed spectra. To avoid the strong absorption of stratosphere, this study analyzed only spectra

collected when solar zenith angle (SZA) was less than 75°.

### 2.3 Spectral retrieval

This study used the QDOAS software developed by Belgian Institute for Space Aeronomy (BIRA-IASB) (http://uv-vis.aeronomie.be/software/QDOAS/, last access: 26 April 2021) to retrieve differential slant column densities (DSCDs) of

the oxygen dimer (O₄), NO₂, HCHO and HONO. A sequential zenith spectrum, estimated from interpolation of two zenith

spectra recorded before and after an elevation sequence, was selected as Frauenhofer reference spectrum. The retrieval

configuring settings followed Xing et al. (2020, 2021), and the configurations used in CINDI intercomparison campaign as

well (Roscoe et al., 2010; Kreher et al., 2019; Wang et al., 2020) (details are listed in Table 1). Fig. 3 illustrates the DOAS

fits of above four species, and reasonably good fitting with tiny values of RMS can be found. Retrieved data with root mean

square (RMS) values larger than $5\times10^{-4}$, $5\times10^{-4}$, $6\times10^{-4}$ and $5\times10^{-3}$ for O₄, NO₂, HCHO and HONO, respectively, were

filtered out. We calculated also the color index (CI), defined as the ratio of spectral intensities at 330 nm to that at 390 nm, to

remove the cloud effects (Wagner et al., 2016). We filtered out data when the CI was less than 10% of the threshold that

obtained through fitting a fifth-order polynomial to CI data (Ryan et al., 2018). After these processes, 90.17%, 86.41%,

83.22% and 80.19%, respectively, of the original DSCDs data were marked as qualified.

### 2.4 Vertical profile retrieval algorithm

The vertical profiles of aerosol extinction and volume mixing ratios (VMR) of trace gases (i. e., NO₂, HCHO and HONO)

were retrieved using the optimal estimation method (OEM) based algorithm. The radiative transfer model VLIDORT (Spurr

et al., 2006) was used as the forward model. The maximum posteriori state vector $x$ was determined through optimizing the

following cost function $\chi^2$.

$$\chi^2 = \left(y - F\left(x,b\right)\right)^T S_\varepsilon^{-1}\left(y - F\left(x,b\right)\right) + \left(x - x_a\right)^T S_a^{-1}\left(x - x_a\right) \quad (1)$$

where $F\left(x,b\right)$ denotes the measurement vector $y$ (DSCDs measured at different elevation angles) as a function of the state

vector $x$ (pollutants profiles) and the true atmospheric meteorological parameters (profiles of temperature and pressure,





albedo and aerosol phase function). $x_a$ stands for a priori state vector. $S_\varepsilon$ and $S_a$ represent the covariance matrices of $y$ and

$x_a$, respectively. During the retrieval process, exponential decreasing shape was assumed as the initial a priori vertical profile shape of aerosol and trace gases, and the corresponding WRF-Chem simulated AOD and VCDs were also used as

input a priori information. The inversion strategy used a Gauss-Newton (GN) scheme (Wedderburn et al., 1974). The iterative weighting function K was calculated using the Jacobians of DSCDs. The inversion consisted of two steps. Aerosol vertical profile was retrieved first, and then fed into the forward model to retrieve trace gases profile.

In this study, we derived the vertical profiles of aerosol and trace gases at 30 vertical layers, covering from 0.0 to 3.0 km, and a 1.0 km correlation length was chosen. For forward simulations, surface albedo and surface altitude were set as 0.08

and 4.2 km, respectively. In addition, a fixed single scattering albedo (SSA) of 0.85 and an aerosol phase function with an asymmetry parameter of 0.65 were selected, due to the low uncertainties in retrieved aerosols with fixed SSA and asymmetry parameter (Irie et al., 2008). The profiles of aerosol and trace gases were filtered out when the degree of freedom (DFS) was less than 1.0 and retrieved relative error were larger than 100%.

### 3 Vertical profile observations of atmospheric composition

Fig. 4 illustrates the monthly data integrity of vertical profiles of aerosol, $NO_2$, HCHO and HONO at the CAS (QOMS). The value of 100 indicates completely continuous observations. Better data integrity occurred mostly in autumn, and serious missing happened mainly in summer. Missing data were largely associated with power outages of the instrument or the measurement station, technical problems of the instrument, poor data quality under unfavorable weather conditions (i.e. rain or high cloud coverage), and retrieval failure (i.e. high retrieval error or low DFS). Considering the heterogeneity of

atmospheric composition in the vertical direction, the resolution of vertical profiles within 0-1.0 km and 1.0-3.0 km were set to 0.1 km and 0.2 km, respectively. During the day, PBL over the TP could reach as high as 1-3km (Chen et al., 2016; Xu et al., 2019). We thus used three representative layers, namely 0-100 m, 500-600 m and 900-1000 m, were used to represent the bottom, middle and upper boundary layers.

### 3.1 Aerosol

As indicated in Fig. 5, maximum AOD over the TP occurred in August (1.19), almost doubled the minimum that happened in April (0.57). However, the high levels of AOD in summer might have been overestimated due to poor integrity of AOD data in summer (Fig. 4: 12.91% in July and 22.58% in August). Relatively enhanced levels of AOD started from September and persisted till February. Such an enrichment was likely to be associated with both the increased anthropogenic emissions due to tourism and transported pollution plume from South Asia. The ground surface of CAS (QOMS) station is exposed

continually throughout the year, rarely covered by ice, snow or vegetation. Valley and mountain winds in this region began



to strengthen from October, and strong winds persisted throughout the whole winter to blow soil and dust particles into the atmosphere (Xu et al., 2015; Liu et al., 2017; Chen et al., 2018).

Fig. 6 (a) shows the time series of hourly averaged aerosol extinction coefficient at layers of 0-100 m, 500-600 m and 900-1000 m December 2017 to February 2019. It suggests that elevated abundance of aerosol was mainly concentrated in the

lower layer, and aerosol loadings decreased with height. Consistent with the seasonality shown in Fig. 5, high aerosol extinction coefficients appeared frequently in autumn and winter. Fig. 6(a) also indicates that strong aerosol extinction coefficients occurred occasionally in the middle layer, associated with long-range transport of particles (Zhang et al., 2018; Chen et al., 2018). The total and seasonal averaged vertical profiles of aerosol extinction are displayed in Fig. 7. All averaged profiles exhibit exponential decreasing shape with maximum values occurred near the surface. Aerosol extinction

coefficient in the lower layer varied with seasons, with maximum in autumn (1.55 km$^{-1}$) and minimum in spring (1.00 km$^{-1}$). The ratios of aerosol extinction coefficients in the middle layer to those in the lower boundary layer were 44.96%, 41.11%, 44.35%, 40.01% and 52.31% for the total-averaged, spring-averaged, summer-averaged, autumn-averaged and winter-averaged aerosol profiles, respectively. These numbers decreased to 24.81%, 22.72%, 25.87%, 21.29% and 30.02% for the upper layer. Relatively higher levels of aerosol above ground during winter is associated with strong mountain-valley breeze,

which can blow dust on the ground surface into high layers (Liu et al., 2017). Aerosol pollution is also more severe in winter in South Asia (Gao et al., 2019), which can be transported as an important source of aerosols over the TP (Cong et al., 2015; Chen et al., 2018; Zhang et al., 2018).

Fig. 8 (a-e) illustrates the diurnal variations of aerosol vertical profiles for different seasons. Lower aerosol extinction mainly occurred before 10:00, and it gradually increased and spread to higher altitudes with the rise of planetary boundary layer

(PBL) height after 10:00. Moreover, aerosol extinction showed bi-peak patterns in all four seasons. One peak appeared between 10:30-12:30, and the other one occurred after 14:00. This bi-peak pattern was in line with previous investigation by Liu et al. (2017), dominated by the effects of local aerosol emissions, local dust geomorphology and mountain valley breeze (Liu et al., 2017). We observed also that the diffusion height of aerosol exhibited maximum in summer and minimum in winter, mainly due to differences in PBL height driven by temperature in four seasons.

**3.2 Nitrogen dioxide (NO$_2$)**

Fig. 6 (b) presents the time series of hourly NO$_2$ concentration at 0-100 m, 500-600 m and 900-1000 m layers from December 2017 to February 2019. Abundance of NO$_2$ in the lower, middle and upper boundary layers exhibited clearly different levels, and elevated levels were concentrated in the lower levels. Higher NO$_2$ concentrations were observed in May and September-December (Chen et al., 2019), which was likely to be associated with the enhanced vehicle emissions due to

arrival of the tourist season in Tibet from May. Additionally, the transmission of NO$_2$ emitted from the Indo-Gangetic Plain (Chen et al., 2017; Yang et al., 2018a) and the enhanced local anthropogenic emissions (i. e. heating using cow dung as fuel) in cold periods also caused the relatively high NO$_2$ levels.



The vertical structure of NO$_2$ presented also an exponential decreasing shape, with the highest concentration located in the lower layer (Fig. 9). NO$_2$ concentrations in the lower layer peaked in autumn (1.28 ppb), and minimum happened in spring (0.82 ppb). The ratios of NO$_2$ concentrations in the middle layer to those in the lower boundary layer were 39.50%, 38.14%, 40.19%, 40.16% and 39.39% for the total-averaged, spring-averaged, summer-averaged, autumn-averaged and winter-averaged aerosol profiles, respectively. These numbers declined to 18.10%, 17.36%, 18.50%, 18.44% and 18.07% for the upper layer. Different from aerosol extinction, higher levels of NO$_2$ above ground occurred mainly in summer, instead of winter, which might be associated with stronger mixing within PBL in summer.

The diurnal variation of NO$_2$ in spring, summer and winter also exhibited a bi-peak pattern (Fig. 8 (f-j)), with one peak within 11:00-13:00 and the other after 16:00. These two periods are usually accompanied with low PBL height. Lower NO$_2$ concentration in the morning (before 10:00) can be attributed to less vehicle emissions (Wang et al., 2011). The elevated PBL height was conducive to the diffusion and dilution of NO$_2$ and lead to lower NO$_2$ concentrations from 13:00 to 16:00. In autumn, another two NO$_2$ peaks values were observed between 14:00-16:00, which was likely to be related to tourist traffic.

**3.3 Formaldehyde (HCHO)**

Formaldehyde (HCHO) is one of the most abundant volatile organic compounds (VOCs) in the atmosphere, and it strongly correlates with peroxy radicals (Kleinman et al., 2001; Ling et al., 2017). HCHO plays a critical role in atmospheric photochemistry to strongly drive ozone (O$_3$) formation (Luecken et al., 2012; Hassan et al., 2018). It comes from both primary sources (i.e. fossil fuel combustion, biomass burning, traffic and industrial activities) and secondary production (i.e. the oxidation of biogenic VOCs). Fig. 6 (c) displays the time series of hourly averaged HCHO concentration at 0-100 m, 500-600 m and 900-1000 m layers from December 2017 to February 2019. We found HCHO at the CAS (QOMS) station was mainly distributed in the lower boundary layer, its concentrations decreased with height, similarly to NO$_2$. The highest HCHO concentrations mainly appeared in September-November, and the lowest HCHO concentrations occurred in December-February. The highest amount of vegetation appeared in autumn at the CAS (QOMS) station, and isoprene along with other active VOCs emitted from vegetation can accumulate near the surface and be oxidized to form high levels of HCHO (Mu et al., 2007). On the other hand, local outdoor biomass burning is active in this season (Chen et al., 2018). The incomplete combustion could emit methane and non-methane volatile organic compounds (NMVOCs) to enhance HCHO concentrations (Gonzi et al., 2011). Fewer primary and secondary sources of HCHO was observed in winter.

Fig. 10 provides the total and seasonal averaged vertical profiles of HCHO. Similarly to NO$_2$, vertical profiles of HCHO in all seasons displayed exponential decreasing shape, with the highest concentration in the lower boundary layer (total averaged value being 3.15 ppb). The HCHO concentrations in the lower layer peaked in autumn (5.20 ppb) and dropped in winter (2.01 ppb). However, the vertical gradient of HCHO was smaller than that of NO$_2$. The ratios of HCHO concentrations in the middle layer to those in the lower boundary layer were 62.86%, 69.74%, 68.06%, 60.00% and 60.20% for the total-averaged, spring-averaged, summer-averaged, autumn-averaged and winter-averaged aerosol profiles,





respectively. These numbers dropped to 42.86%, 47.60%, 46.27%, 40.77% and 41.14% for the upper layer. The vertical gradient of HCHO in autumn was relatively larger than that in other three seasons, which was partially associated with stronger surface HCHO sources in autumn.

On the diurnal variations of vertical profiles of HCHO, lower concentrations were observed before 10:00, but the abundance gradually increased with sunrise and elevation of temperature (Fig. 8 (k-o)). HCHO peaked within 10:00-16:00 in spring and
winter, while peak values of HCHO also appeared after 16:00 due to prolonged daytime in summer and autumn. The maximum and minimum diffusion heights of HCHO appeared in summer and winter, respectively, associated with PBL height.

### 3.4 Nitrous acid (HONO)

The photolysis of nitrous acid (HONO) is a significant source of hydroxyl (OH) radical. HONO mainly originates from
primary emissions from vehicles, ships, biomass burning and soil, the photolysis of nitrate particles ($NO_3^-$ ),homogeneous reaction of NO with OH radical, and heterogeneous reaction of $NO_2$ on several types of surfaces (Wang et al., 2015; Fu et al., 2019). As shown in Fig. 6 (d), elevated HONO was mostly distributed in the lower boundary layer, and higher HONO concentrations appeared from July 2018 to February 2019. Enhanced levels after summer was likely to be connected with elevated direct emissions from vehicle, biomass burning and soil (Wang et al., 2015; Fu et al., 2019). We found high aerosol
extinction, $NO_2$ and HONO concentrations simultaneously in Fig. 8 (d), (i) and (s), suggesting that heterogeneous reaction of $NO_2$ on aerosol surfaces might be another important source of HONO over the TP.

Fig. 11 shows that the vertical profiles of HONO in all seasons exhibited exponential shape, with the highest concentration (0.93 ppb) in the lower boundary layer. HONO concentrations in the lower layer peaked in summer (1.11 ppb) and dropped in spring (0.56 ppb) (Fig. 11). Different from aerosol, $NO_2$ and HCHO, HONO concentrations in the middle and upper layers
were extremely low. The proportions of HCHO concentrations in the middle layer to those in the lower boundary layer were 8.86%, 13.68%, 7.82%, 9.05% and 10.44% for the total-averaged, spring-averaged, summer-averaged, autumn-averaged and winter-averaged aerosol profiles, respectively. These numbers decreased to 3.14%, 5.30%, 3.52%, 2.72% and 3.06% for the upper layer. On the diurnal variations of vertical profiles of HONO, we found that HONO concentrations were, consistently in four seasons, lower before 10:00 and peaked around 10:00-14:00. Such as pattern in the monitoring site was different
from that in other low altitude cities (Wang et al., 2013; Lee et al., 2016; Wang et al., 2017; Xing et al., 2021).

### 3.5 Validations against satellite retrievals

In this study, OMI $NO_2$ and HCHO products were used to validate the described dataset. For better comparison, MAX-DOAS data were averaged over the OMI overpass time (from 06:30 to 08:30 UTC) in the region that covers the CAS (QOMS) station. OMI data were spatially averaged over the 15 km grid cells around the MAX-DOAS site, as the spatial
resolution of OMI is 15×15 km². The VCDs of $NO_2$ and HCHO were derived by vertical integrations of their vertical

profiles. The linear association between MAX-DOAS and OMI measurements are shown in Fig. 12. We found reasonably good correlations between MAX-DOAS and OMI observations, with Pearson correlation coefficients (R) of 0.68 and 0.60 (slope of 1.21 and 1.12, offset of $1.29\times10^{15}$ and $2.06\times10^{15}$ molec. cm$^{-2}$) for $NO_2$ and HCHO, respectively. OMI $NO_2$ and HCHO VCDs were systematically lower than MAX-DOAS $NO_2$ and HCHO VCDs, which is different from previous

comparisons applied in lower-altitude areas (Liu et al., 2016; Xing et al., 2017). This might be caused by large-area averaging effect of OMI and introduced uncertainties in OMI products by cloud cover and high surface albedo.

We observed also that the correlations between MAX-DOAS and OMI observations exhibited significant differences across seasons. The highest correlation appeared in autumn, followed by winter (Table S1). High correlation in these two seasons was associated with increased tourism vehicle emissions, regional transport of $NO_2$, and enhanced photochemical formation

of HCHO. Failed linear regression analysis mainly appeared in spring and summer, due to high cloud coverage from March to September in the TP (Chen et al., 2006).

## 4 Data availability

Ground-based vertical profile observations of atmospheric composition on the Tibetan Plateau introduced here are available in .txt format at Zenodo (http://doi.org/10.5281/zenodo.4911384; Xing et al., 2021).

## 5 Summary

In situ observations, especially profile observations, are scarce but essential in polar regions, including the TP, the so-called "Third Pole". The sources of atmospheric composition over the TP, and how the species interact with regional and global climate remain unclear, mainly owing to lack of observations. Large uncertainties also remain in chemical transport and climate modeling of cold-region processes, without observational constraints. The presented dataset offers high time-

resolution vertical profile observations of aerosol, $NO_2$, HCHO and HONO for more than one year on the TP. Distinct features were found on the diurnal variations, vertical distributions, and seasonality of these atmospheric composition species. Data products were also validated with satellite retrievals.

This dataset provides better temporal coverage and fills the gaps in the vertical direction. It has potentials to improve both satellite retrievals and reduce uncertainties in chemical transport modeling over the TP, particularly in the vertical direction.

We expect this dataset can be used to understand the sources and dynamic evolution of air pollutants over the TP, to assess the impacts of chemical forcers on climate system at multiple scales, and to facilitate the development and improvement of models in cold regions.



*Author contributions.* All the authors were involved in the generation of the introduced dataset. Chengzhi Xing, Cheng Liu
and Meng Gao wrote the manuscript with contributions from all the other authors.

*Competing interests.* The authors declare that they have no conflict of interest

*Acknowledgements.* We would like to thank the Institute of Tibetan Plateau Research, Chinese Academy of Sciences
(Professor Zhiyuan Cong's group) for help on experimental setup at the CAS monitoring station.

*Financial support.* This research is supported by grants from the National Key Research and Development Program of China
(No. 2018YFC0213104), National Natural Science Foundation of China (No.41722501, 91544212, 51778596, 41575021,
41977184 and 41875043), National Key Research and Development Program of China (No.2017YFC0210002,
2016YFC0203302 and 2017YFC0212800), Anhui Science and Technology Major Project (No.18030801111), the Strategic
Priority Research Program of the Chinese Academy of Sciences (No. XDA23020301), the National Key Project for Causes
and Control of Heavy Air Pollution (No.DQGG0102 and DQGG0205), the National High-Resolution Earth Observation
Project of China (No.05-Y30B01-9001-19/20-3) and Civil Aerospace Technology Advance Research Project (No.
Y7K00100KJ).

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

Kreher, Karin, Roozendael, Michel Van, Hendrick, Francois, Apituley, Arnoud, Dimitropoulou, Ermioni, Frieß, Udo,
Richter, Andreas,Wagner, Thomas, Abuhassan, Nader, Ang, Li, Anguas, Monica, Bais, Alkis, Benavent, Nuria, Bösch, Tim,
Bognar, Kristof, Borovski, Alexander, Bruchkouski, Ilya, Cede, Alexander, Ka, L. Chan, Donner, Sebastian, Drosoglou,
Theano, Fayt, Caroline, Finkenzeller, Henning, Garcia-Nieto, David, Gielen, Clio, Gómez-Martín, Laura, Hao, Nan,
Herman, Jay R., Hermans, Christian, Hoque, Syedul, Irie, Hitoshi, Jin, Junli, Johnston, Paul, Butt, Junaid Khayyam,
Khokhar, Fahim, Koenig, Theodore K., Kuhn, Jonas, Kumar, Vinod, Lampel, Johannes, Liu, Cheng, Ma, Jianzhong,
Merlaud, Alexis, Mishra, Abhishek K., Müller, Moritz, Navarro-Comas, Monica, Ostendorf, Mareike, Pazmino, Andrea,
Peters, Enno, Pinardi, Gaia, Pinharanda, Manuel, Piters, Ankie, Platt, Ulrich, Postylyakov, Oleg, Prados-Roman, Cristina,
Puentedura, Olga, Querel, Richard, Saiz-Lopez, Alfonso, Schönhardt, Anja, Schreier, Stefan F., Seyler, Andre, Sinha,
Vinayak, Spinei, Elena, Strong, Kimberly, Tack, Frederik, Tian, Xin, Tiefengraber, Martin, Tirpitz, Jan-Lukas, van Gent,
Jeron, Volkamer, Rainer, Vrekoussis, Mihalis, Wang, Shanshan, Wang, Zhuoru, Wenig, Mark, Wittrock, Folkard, Xie,
Pinhua H., Jin, Xu, Yela, Margarita, Zhang, Chengxin, and Zhao, Xiaoyi: Intercomparison of NO2, O4, O3 and HCHO slant





column measurements by MAX-DOAS and zenith-sky UV-Visible spectrometers during the CINDI-2 campaign, Atmos. Meas. Tech. Disc., 2019.

Lau, W. K., Kim, M. K., Kim, K. M., and Lee, W. S.: Enhanced surface warming and accelerated snow melt in the Himalayas and Tibetan Plateau induced by absorbing aerosols, Environ. Res. Lett., 5(2), p.025204, 2010.

Lee, J. D., Whalley, L. K., Heard, D. E., Stone, D., Dunmore, R. E., Hamilton, J. F., Young, D. E., Allan, J. D., Laufs, S., and Kleffmann, J.: Detailed budget analysis of HONO in central London reveals a missing daytime source, Atmos. Chem. Phys. 16, 2747–2764, 2016.

Li, C., Bosch, C., Kang, S., Andersson, A., Chen, P., Zhang, Q., Cong, Z., Chen, B., Qin, D., and Gustafsson, O.: Sources of black carbon to the Himalayan-Tibetan Plateau glaciers, Nat. Commun., 7, 12574, 2016.

Li, R., Zhao, Y., Zhou, W., Meng, Y., Zhang, Z., and Fu, H.: Developing a novel hybrid model for the estimation of surface 8 h ozone (O3) across the remote Tibetan Plateau during 2005–2018, Atmos. Chem. Phys., 20, 6159–6175, 2020.

Liang, F., Gao, M., Xiao, Q., Carmichael, G.R., Pan, X. and Liu, Y.: Evaluation of a data fusion approach to estimate daily
PM2.5 levels in North China, Env. Res., 158, 54-60, https://doi.org/10.1016/j.envres.2017.06.001, 2017.

Ling, Z. H., Zhao, J., Fan, S. J., and Wang, X. M.: Sources of formaldehyde and their contributions to photochemical O3 formation at an urban site in the Pearl River Delta, southern China, Chemosphere, 168: 1293-301, 2017.

Liu, B., Cong, Z., Wang, Y., Xin, J., Wan, X., Pan, Y., Liu, Z., Wang, Y., Zhang, G., Wang, Z., Wang, Y., and Kang, S.: Background aerosol over the Himalayas and Tibetan Plateau: observed characteristics of aerosol mass loading, Atmos.
Chem. Phys., 17, 449–463, 2017.

Liu, C., Gao, M., Hu, Q., Brasseur, G. P., and Carmichael, G. R.: Stereoscopic Monitoring: A Promising Strategy to Advance Diagnostic and Prediction of Air Pollution, Bull. Am. Meteorol. Soc., E730-E737, 2021a.

Liu, C., Xing, C., Hu, Q., Li, Q., Liu, H., Hong, Q., Tan, W., Ji, X., Lin, H., Lu, C., Lin, J., Liu, H., Wei, S., Chen, J., Yang, K., Wang, S., Liu, T., and Chen, Y.: Ground-based hyperspectral stereoscopic remote sensing network: A promising strategy
to learn coordinated control of O3 and PM2.5 over China, Engineering, 2021b.

Liu, H., Liu, C., Xie, Z., Li, Y., Huang, X., Wang, S., Xu, J., and Xie, P.: A paradox for air pollution controlling in China revealed by "APEC Blue" and "Parade Blue", Sci. Rep., 6, 34408, 2016.

Liu, H., Wang, Q., Xing, L., Zhang, Y., Zhang, T., Ran, W., and Cao, J.: Measurement report: quantifying source contribution of fossil fuels and biomass-burning black carbon aerosol in the southeastern margin of the Tibetan Plateau,
Atmos. Chem. Phys., 21, 973–987, 2021.

Liu, Y., Hoskins, B., and Blackburn, M.: Impact of Tibetan Orography and Heating on the Summer Flow over Asia, J. Meteorol. Soc. Jpn. Ser. II, 85B, 1–19, 2007.

Luecken, D. J., W. T. Hutzell, M. L. Strum, and G. A. Pouliot: Regional sources of atmospheric formaldehyde and acetaldehyde, and implications for atmospheric modeling, Atmos. Environ., 47: 477-90, 2012.

Lu, Z., Streets, D. G., Zhang, Q., and Wang, S.: A novel back trajectory analysis of the origin of black carbon transported to the Himalayas and Tibetan Plateau during 1996–2010, Geophys. Res. Lett., 39, L01809, 2012.



Ma, Y., Hu, Z., Xie, Z., Ma, W., Wang, B., Chen, X., Li, M., Zhong, L., Sun, F., Gu, L., Han, C., Zhang, L., Liu, X., Ding, Z., Sun, G., Wang, S., Wang, Y., and Wang, Z.: A long-term (2005–2016) dataset of integrated land-atmosphere interaction observations on the Tibetan Plateau, V1, Science Data Bank, 2020.

Ma, Y., Hu, Z., Xie, Z., Ma, W., Wang, B., Chen, X., Li, M., Zhong, L., Sun, F., Gu, L., Han, C., Zhang, L., Liu, X., Ding, Z., Sun, G., Wang, S., Wang, Y., and Wang, Z.: A long-term (2005–2016) dataset of hourly integrated land–atmosphere interaction observations on the Tibetan Plateau, Earth Syst. Sci. Data, 12, 2937–2957, 2020.

Meller, R., and Moortgat, G.K.: Temperature dependence of the absorption cross sections of formaldehyde between 223 and 323K in the wavelength range 225–375 nm, J. Geophys. Res. 105, 7089–7101, 2000.

Mu, Y., Pang, X., Quan, J., and Zhang, X.: Atmospheric carbonyl compounds in Chinese background area: A remote mountain of the Qinghai-Tibetan Plateau, J. Geophys. Res.-Atmos., 112, D22302, 2007.

Peng, J., Liu, Z., Liu, Y., Wu, J. and Han, Y.: Trend analysis of vegetation dynamics in Qinghai–Tibet Plateau using Hurst Exponent, Ecol. Indic., 14(1), 28-39, 2012.

Pu, Z. X., Xu, L., and Salomonson, V. V.: MODIS/Terra observed seasonal variations of snow cover over the Tibetan 485 Plateau, Geophys. Res. Lett., 34, 2007.

Qiu, J.: China: the third pole, Nature, 454, 393–396, 2008.

Ran, L., Lin, W. L., Deji, Y. Z., La, B., Tsering, P. M., Xu, X. B., and Wang, W.: Surface gas pollutants in Lhasa, a highland city of Tibet – current levels and pollution implications, Atmos. Chem. Phys., 14, 10721-10730, 2014.

Robert J. D. Spurr.: VLIDORT: A linearized pseudo-spherical vector discrete ordinate radiative transfer code for forward 490 model and retrieval studies in multilayer multiple scattering media, J. Quant. Spectrosc. RA., 102, 316-342, 2006.

Roscoe, H. K., Van Roozendael, M., Fayt, C., du Piesanie, A., Abuhassan, N., Adams, C., Akrami, M., Cede, A., Chong, J., Clémer, K., Friess, U., Gil Ojeda, M., Goutail, F., Graves, R., Griesfeller, A., Grossmann, K., Hemerijckx, G., Hendrick, F., Herman, J., Hermans, C., Irie, H., Johnston, P.V., Kanaya, Y., Kreher, K., Leigh, R., Merlaud, A., Mount, G.H., Navarro, M., Oetjen, H., Pazmino, A., Perez-Camacho, M., Peters, E., Pinardi, G., Puentedura, O., Richter, A., Schönhardt, A., 495 Shaiganfar, R., Spinei, E., Strong, K., Takashima, H., Vlemmix, T., Vrekoussis, M., Wagner, T., Wittrock, F., Yela, M., Yilmaz, S., Boersma, F., Hains, J., Kroon, M., Piters, A., and Kim, Y. J.: Intercomparison of slant column measurements of NO2 and O4 by MAX-DOAS and zenith-sky UV and visible spectrometers, Atmos. Meas. Tech., 3, 1629–1646, 2010.

Ramanathan, V., Ramana, M. V., Roberts, G., Kim, D., Corrigan, C. E., Chung, C. E., and Winker, D.: Warming trends in Asia amplified by brow cloud solar absorption, Nature, 448, 575–8, 2007.

Rothman, L. S., Gordon, I. E., Barbe, A., Benner, D. C., Bernath, P. F., Birk, M., Boudon, V., Brown, L. R., Campargue, A., Champion, J. P., Chance, K., Coudert, L. H., Dana, V., Devi, V. M., Fally, S., Flaud, J. M., Gamache, R. R., Goldman, A., Jacquemart, D., Kleiner, I., Lacome, N., Lafferty, W. J., Mandin, J. Y., Massie, S. T., Mikhailenko, S. N., Miller, C. E., Moazzen-Ahmadi, N., Naumenko, O. V., Nikitin, A. V., Orphal, J., Perevalov, V. I., Perrin, A., Predoi-Cross, A., Rinsland, C. P., Rotger, M., Šimečkova̕, M., Smith, M. A. H., Sung, K., Tashkun, S. A., Tennyson, J., Toth, R. A., Vandaele, A. C.,





and Vander Auwera, J.: The HITRAN 2008 molecular spectroscopic database, J. Quant. Spectrosc. Radiat. Transf., 110 (9–10), 533–572, 2010.

Ryan, R. G., Rhodes, S., Tully, M., Wilson, S., Jones, N., and Frieß, U.: Daytime HONO, NO$_{2b}$/N and aerosol distributions from MAX-DOAS observations in Melbourne, Atmos. Chem. Phys. Discuss., 1–27, 2018.

Seidel, D. J., Ao, C. O., and Li, K.: Estimating climatological planetary boundary layer heights from radiosonde
observations: Comparison of methods and uncertainty analysis, J. Geophys. Res., 115, D16113, 2010.

Serdyuchenko, A., Gorshelev, V., Weber, M., Chehade, W., and Burrows, J. P.: High spectral resolution ozone absorption cross-sections – part 2: temperature dependence, Atmos. Meas. Tech., 7, 625–636, 2014.

Shrestha, A. B., Wake, C. P., Mayewski, P. A., and Dibb, J. E.: Maximum temperature trends in the Himalaya and its vicinity: an analysis based on temperature records from Nepal for the period 1971–94, J. Climate, 12(9), 2775-2786, 1999.

Skerlak, B., Sprenger, M., and Wernli, H.: A global climatology of stratosphere–troposphere exchange using the ERA Interim data set from 1979 to 2011, Atmos. Chem. Phys., 14, 913–917, 2014.

Stout, J. E.: Diurnal patterns of blowing sand, Earth Surf. Proc. Land., 35, 314–318, 2010.

Stutz, J., Kim, E. S., Platt, U., Bruno, P., Perrino, C., and Febo, A.: UV–visible absorption cross sections of nitrous acid, J. Geophys. Res.-Atmos. 105, 14585–14592, 2000.

Thalman, R.M., and Volkamer, R.: Temperature dependent absorption cross-sections of O$_2$-O$_2$ collision pairs between 340 and 630 nm and at atmospherically relevant pressure, Phys. Chem. Chem. Phys. 15, 15371–15381, 2013.

Tian, L., Yao, T., MacClune, K., White, J. W. C., Schilla, A., Vaughn, B., Vachon, R., and Ichiyanagi, K.: Stable isotopic variations in west China: A consideration of moisture sources, J. Geophys. Res., 112, D10112, 2007.

Vandaele, A. C., Hermans, C., Simon, P. C., Carleer, M., Colin, R., Fally, S., Mérienne, M.-F., Jenouvrier, A., and Coquart,
B.: Measurements of the NO2 absorption cross section from 42 000 cm−1 to 10000 cm−1 (238–1000 nm) at 220 K and 294 K, J. Quant. Spectrosc. Ra., 59, 171–184, 1998.

Vlemmix, T., Piters, A. J. M., Berkhout, A. J. C., Gast, L. F. L., Wang, P., and Levelt, P. F.: Ability of the MAX-DOAS method to derive profile information for NO2: can the boundary layer and free troposphere be separated? Atmos. Meas. Tech., 4: 2659–2684, 2011.

Wang, K., Hattori, S., Lin, M., Ishino, S., Alexander, B., Kamezaki, K., Yoshida, N., and Kang, S.: Isotopic constraints on atmospheric sulfate formation pathways in the Mt. Everest region, southern Tibetan Plateau, Atmos. Chem. Phys., 21, 8357–8376, 2021.

Wang, S., Zhou, R., Zhao, H., Wang, Z., Chen, L., and Zhou, B.: Long-term observation of atmospheric nitrous acid (HONO) and its implication to local NO2 levels in Shanghai, China, Atmos. Environ., 77, 718–724, 2013.

Wagner, T., Beirle, S., Remmers, J., Shaiganfar, R., and Wang, Y.: Absolute calibration of the colour index and O4 absorption derived from multi AXis (MAX-)DOAS measurements and their application to a standardised cloud classification algorithm, Atmos. Meas. Tech., 9, 4803–4823, 2016.





Wang, F., Lin, W., Wang, J., and Zhu, T.: NOx release from snow and ice covered surface in polar regions and the Tibetan Plateau, Advances in Climate Change Research, 2(3), 141-148, 2011.

Wang, Y., Arnoud, A., Alkiviadis, B., Steffen, B., Nuria, B., Alexander, B., Ilya, B., Chan, K. L., Sebastian, D., Theano, D., Henning, F., Martina, M. F., Udo F., David, G. N., Laura, G. M., François, H., Andreas, H., Jin, J., Paul, J., Theodore, K. K., Karin, K., Vinod, K., Aleksandra, K., Johannes, L., Liu, C., Liu, H., Ma, J., Oleg, L. P., Oleg, P., Richard, Q., Alfonso, S. L., Stefan, S., Tian, X., Tirpitz, J., Roozendael, M. V., Volkamer, R., Wang, Z., Xie, P., Xing, C., Xu, J., Margarita, Y., Zhang, C., and Wagner, T.: Inter-comparison of MAX-DOAS measurements of tropospheric HONO slant column densities and

vertical profiles during the CINDI-2 Campaign, Atmos. Meas. Tech., 13, 5087–5116, 2020.

Wang, Y., Beirle, S., Hendrick, F., Hilboll, A., Jin, J., Kyuberis, A. A., Lampel, J., Li, A., Luo, Y., Lodi, L., Ma, J., Navarro, M., Ortega, I., Peters, E., Polyansky, O. L., Remmers, J., Richter, A., Puentedura, O., Roozendael, M. V., Seyler, A., Tennyson, J., Volkamer, R., Xie, P., Zobov, N. F., and Wagner, T.: MAX-DOAS measurements of HONO slant column densities during the MAD-CAT campaign: inter-comparison, sensitivity studies on spectral analysis settings, and error

budget, Atmos. Meas. Tech., 10, 3719–3742, 2017.

Wang, Z., Chan., K. L., Heue, K. P., Doicu, A., Wagner, T., Holla, R., and Wiegner, M.: A multi-axis differential optical absorption spectroscopy aerosol profile retrieval algorithm for high-altitude measurements: application to measurements at Schneefernerhaus (UFS), Germany, Atmos. Meas. Tech., 13, 1835–1866, 2020.

Wedderburn, R. W. M.: Quasi-likelihood functions, generalized linear models, and the Gauss—Newton method, Biometrika,

555  61, 3, 439, 1974.

Wu, G., Liu, Y., He, B., Bao, Q., Duan, A., and Jin, F.-F.: Thermal Controls on the Asian Summer Monsoon, Sci. Rep., 2, 404, 2012.

Wu, G., Liu, Y., Zhang, Q., Duan, A., Wang, T., Wan, R., Liu, X., Li, W., Wang, Z., and Liang, X.: The Influence of Mechanical and Thermal Forcing by the Tibetan Plateau on Asian Climate, J. Hydrometeorol., 8, 770–789, 2007.

Wu, S., Dai, G., Song, X., Liu, B., and Liu, L.: Observations of water vapor mixing ratio profile and flux in the Tibetan Plateau based on the lidar technique, Atmos. Meas. Tech., 9, 1399–1413, 2016.

Xia, X. G., Zong, X. M., Cong, Z. Y., Chen, H. B., Kang, S. C., and Wang, P. C.: Baseline continental aerosol over the central Tibetan plateau and a case study of aerosol transport from South Asia, Atmos. Environ., 45, 7370–7378, 2011.

Xing, C., Liu, C., Hu, Q., Fu, Q., Wang, S., Lin, Hua, Zhu, Y., Wang, S., Wang, W., Javed, Z., Ji, X., and Liu, J.: Vertical

distributions of wintertime atmospheric nitrogenous compounds and the corresponding OH radicals production in Leshan, southwest China, J. Environ. Sci., 105, 44–55, 2021.

Xing, C., Liu, C., Hu, Q., Fu, Q., Lin, H., Wang, S., Su, W., Wang, W., Javed, Z., and Liu, J.: Identifying the wintertime sources of volatile organic compounds (VOCs) from MAX-DOAS measured formaldehyde and glyoxal in Chongqing, southwest China, Sci. Total Environ., 715, 136258, 2020.



Xing, C., Liu, C., Wang, S., Chan, K. L., Gao, Y., Huang, X., Su, W., Zhang, C., Dong, Y., Fan, G., Zhang, T., Chen, Z., Hu, Q., Su, H., Xie, Z., and Liu, J.: Observations of the vertical distributions of summertime atmospheric pollutants and the corresponding ozone production in Shanghai, China, Atmos. Chem. Phys., 17, 14275–14289, 2017.

Xing, C.: Ground-based vertical observations of atmospheric composition from field campaign on the Tibetan Plateau, Zenodo, http://doi.org/10.5281/zenodo.4911384, 2021.

Xu, B., Cao, J., Hansen, J., Yao, T., Joswia, D. R., Wang, N., Wu, G., Wang, M., Zhao, H., Yang, W., Liu, X., and He, J.: Black soot and the survival of Tibetan glaciers, Proc. Natl. Acad. Sci. U.S.A., 106(52), 22,114–22,118, doi:10.1073/pnas.0910444106, 2009.

Xu, C., Ma, Y., You, C., and Zhu, Z.: The regional distribution characteristics of aerosol optical depth over the Tibetan Plateau, Atmos. Chem. Phys., 15, 12065–12078, 2015.

Xu, X. D., Lu, C. G., Shi, X. H., and Gao, S. T.: World water tower: An atmospheric perspective, Geophys. Res. Lett., 35, 2008.

Xu, L., Liu, H., Du, Q. and Xu, X.: The assessment of the planetary boundary layer schemes in WRF over the central Tibetan Plateau. Atmospheric Research, 230, 104644, 2019.

Xu, Y., Ramanathan, V., and Washington, W. M.: Observed high-altitude warming and snow cover retreat over Tibet and 585 the Himalayas enhanced by black carbon aerosols, Atmos. Chem. Phys., 16(3), 1303-1315, 2016.

Yanai, M., Li, C., and Song, Z.: Seasonal Heating of the Tibetan Plateau and Its Effects on the Evolution of the Asian Summer Monsoon, J. Meteorol. Soc. Jpn. Ser. II, 70, 319–351, 1992.

Yang, J. H., Kang, S.C., Ji, Z. M. and Chen, D. L.: Modeling the origin of anthropogenic black carbon and its climatic effect over the Tibetan Plateau and surrounding regions, J. Geophys. Res. 123: 671–692, 2018a.

Yang, J., Kang, S., and Ji, Z.: Sensitivity analysis of chemical mechanisms in the WRF-chem model in reconstructing aerosol concentrations and optical properties in the Tibetan plateau, Aerosol Air Qual. Res., 18, 505–521, https://doi.org/10.4209/aaqr.2017.05.0156, 2018b.

Yang, K., Koike, T., and Yang, D.: Surface flux parameterization in the Tibetan Plateau, Bound. Layer Meteor., 106, 245–262, 2003

Yao, T. D., Thompson, L., Yang, W., Yu, W. S., Gao, Y., Guo, X. J., Yang, X. X., Duan, K. Q., Zhao, H. B., Xu, B. Q., Pu, J. C., Lu, A. X., Xiang, Y., Kattel, D. B., and Joswiak, D.: Different glacier status with atmospheric circulations in Tibetan Plateau and surroundings, Nat. Clim. Change, 2, 663-667, 2012.

Zhang, J., Xia, X., and Wu, X.: First in situ UV profile across the UTLS accompanied by ozone measurement over the Tibetan Plateau, J. Environ., Sci., 98, 71-76, 2020.

Zhang, R., Wang, H., Qian, Y., Rasch, P. J., Easter, R. C., Ma, P. L., Singh, B., Huang, J., and Fu, Q.: Quantifying sources, transport, deposition, and radiative forcing of black carbon over the Himalayas and Tibetan Plateau, Atmos. Chem. Phys., 15, 6205-6223, 2015.



Zhang, X., Xu, J., Kang, S., Liu, Y., and Zhang, Q.: Chemical characterization of long-range transport biomass burning emissions to the Himalayas: insights from high-resolution aerosol mass spectrometry, Atmos. Chem. Phys., 18, 4617–4638, 2018.

Zhu, J., Xia, X., Che, H., Wang, J., Cong, Z., Zhao, T., Kang, S., Zhang, X., Yu, X., and Zhang, Y.: Spatiotemporal variation of aerosol and potential long-range transport impact over the Tibetan Plateau, China, Atmos. Chem. Phys., 19, 14637–14656, 2019.

Zou, H., and Gao, Y.: Vertical ozone profile over Tibet using sage I and II data, Adv. Atmos. Sci., 14, 505-512, 1997.













**Table 1. Detailed retrieval settings of O4, NO2, HCHO and HONO.**

| Parameter | Data source | Fitting intervals (nm) | | | |
|---|---|---|---|---|---|
| | | $O_4$ | $NO_2$ | HCHO | HONO |
| Wavelength range | | 338-370 | 338-370 | 322.5-358 | 335-373 |
| $NO_2$ | 298K, $I_0$-corrected, Vandaele et al. (1998) | ✓ | ✓ | ✓ | ✓ |
| $NO_2$ | 220K, $I_0$-corrected, Vandaele et al. (1998) | ✓ | ✓ | ✗ | ✓ |
| $O_3$ | 223K, $I_0$-corrected, Serdyuchenko et al. (2014) | ✓ | ✓ | ✓ | ✓ |
| $O_3$ | 243K, $I_0$-corrected, Serdyuchenko et al. (2014) | ✓ | ✓ | ✓ | ✓ |
| $O_4$ | 293K, Thalman and Volkamer (2013) | ✓ | ✓ | ✓ | ✓ |
| HCHO | 298K, Meller and Moortgat (2000) | ✓ | ✓ | ✓ | ✓ |
| $H_2O$ | HITEMP (Rothman et al. 2010) | ✓ | ✓ | ✗ | ✓ |
| BrO | 223K, Fleischmann et al. (2004) | ✓ | ✓ | ✓ | ✓ |
| HONO | 296K, Stutz et al. (2000) | ✗ | ✗ | ✗ | ✓ |
| Ring | Calculated with QDOAS | ✓ | ✓ | ✓ | ✓ |
| Polynomial degree | | Order 3 | Order 3 | Order 5 | Order 5 |
| Intensity offset | | Constant | Constant | Constant | Constant |

\* Solar $I_0$ correction; Aliwell et al. (2002)

Earth System
Science
Data

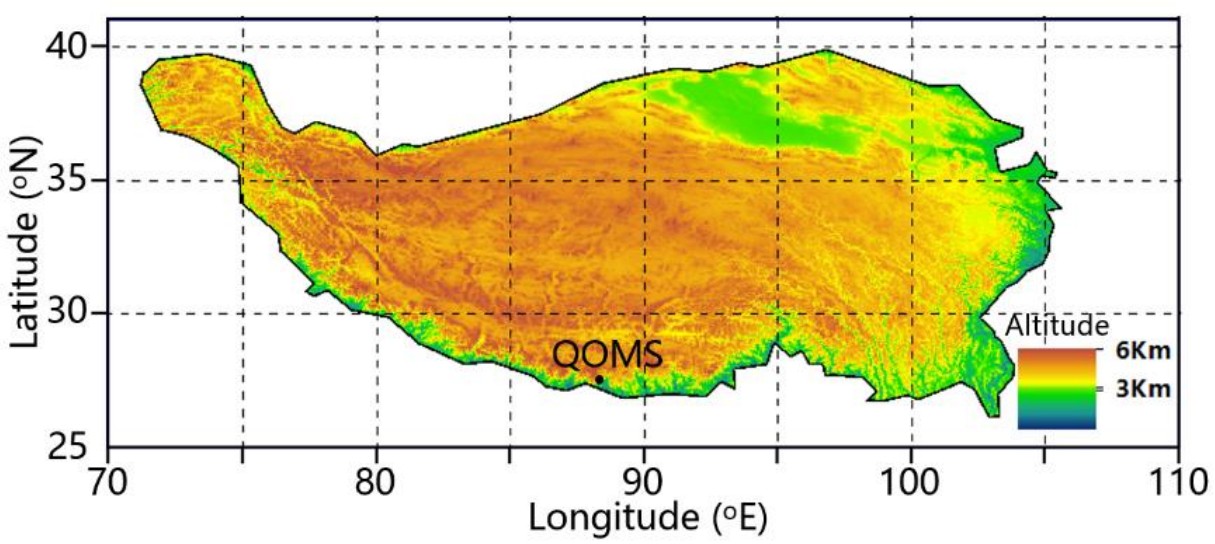

**645** **Figure 1. Elevation map of the TP and location of the CAS (QOMS) monitoring site.**

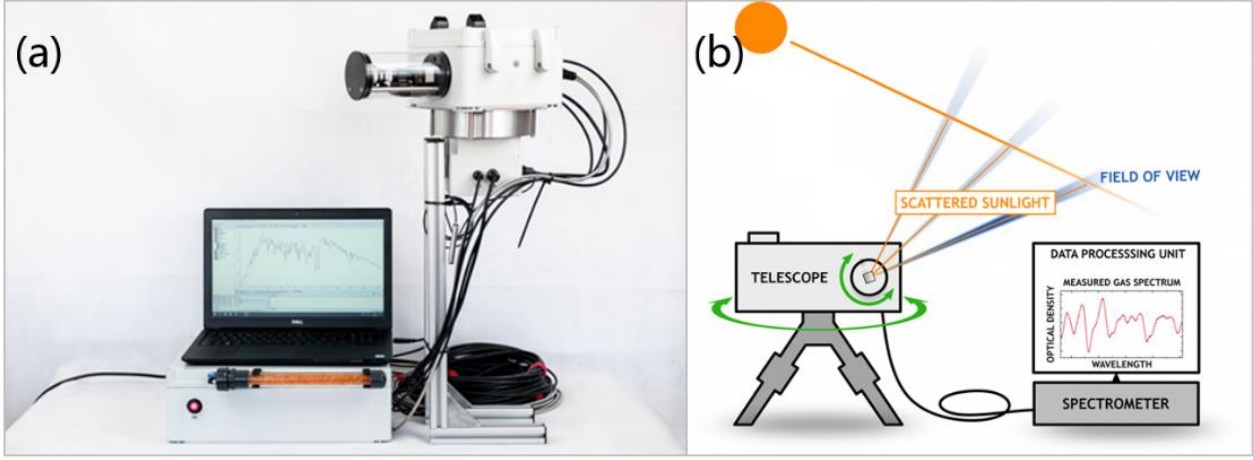

**Figure 2. (a) Physical view and (b) schematic view of the MAX-DOAS instrument.**



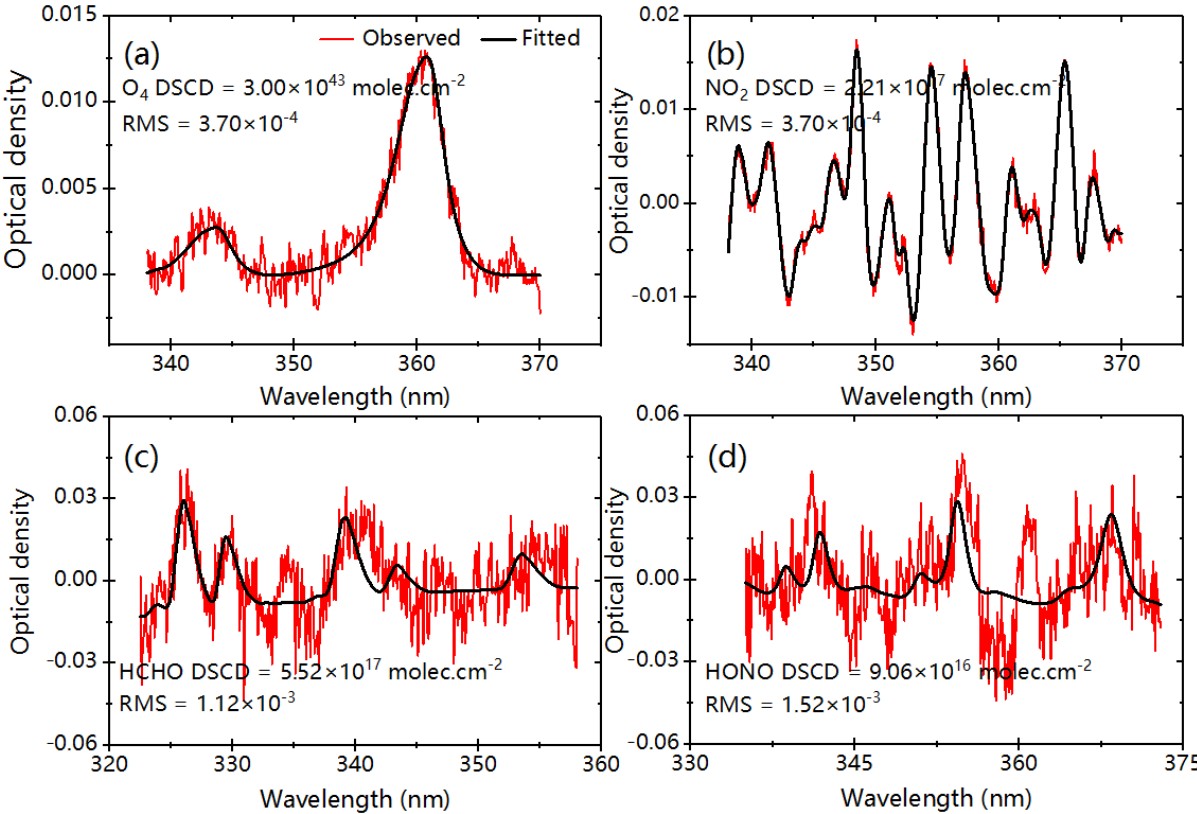

**Figure 3. Examples of DOAS fits for O4, NO2, HCHO and HONO (~13:30 LST on May 22, 2018, SZA and EVA values were 52.53o and 15o, respectively).**





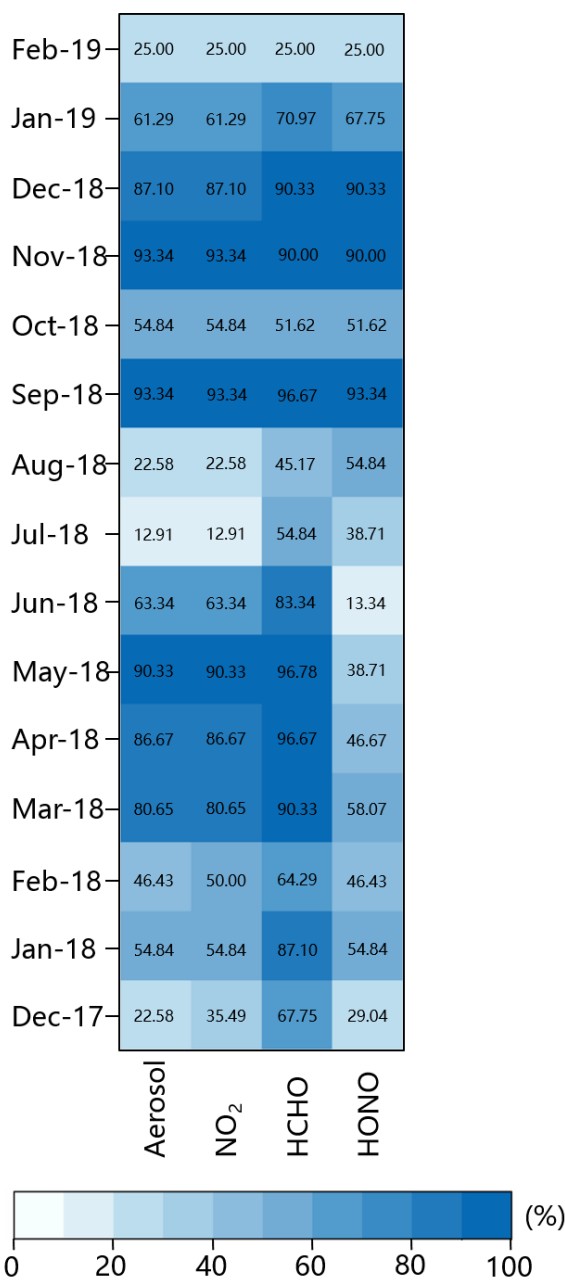

Figure 4. Monthly data integrity of the vertical profiles of aerosol, NO2, HCHO and HONO.






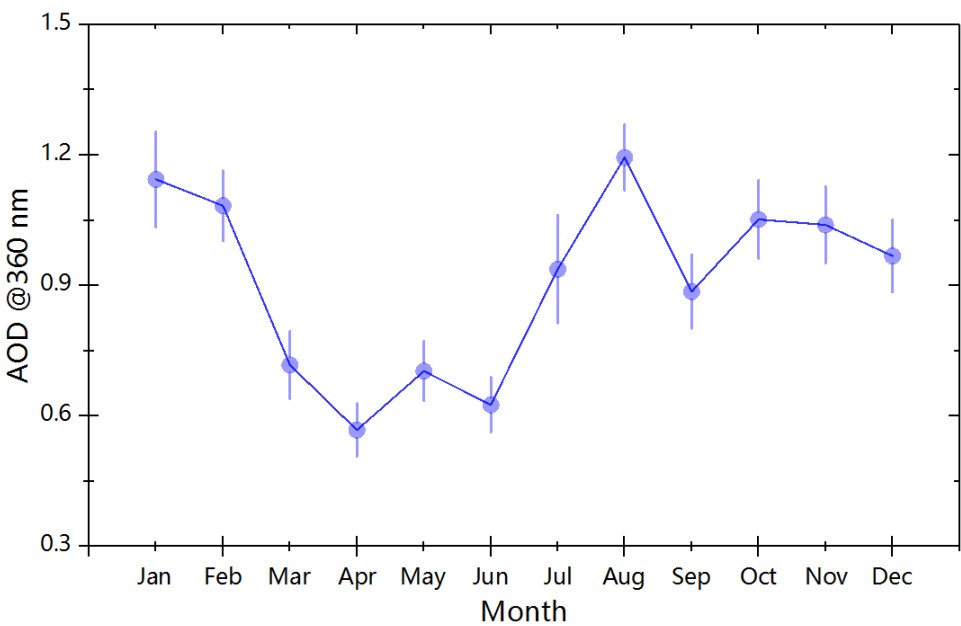

**Figure 5. Time series of monthly AOD from December 2017 to February 2019 at the CAS (QOMS) station.**

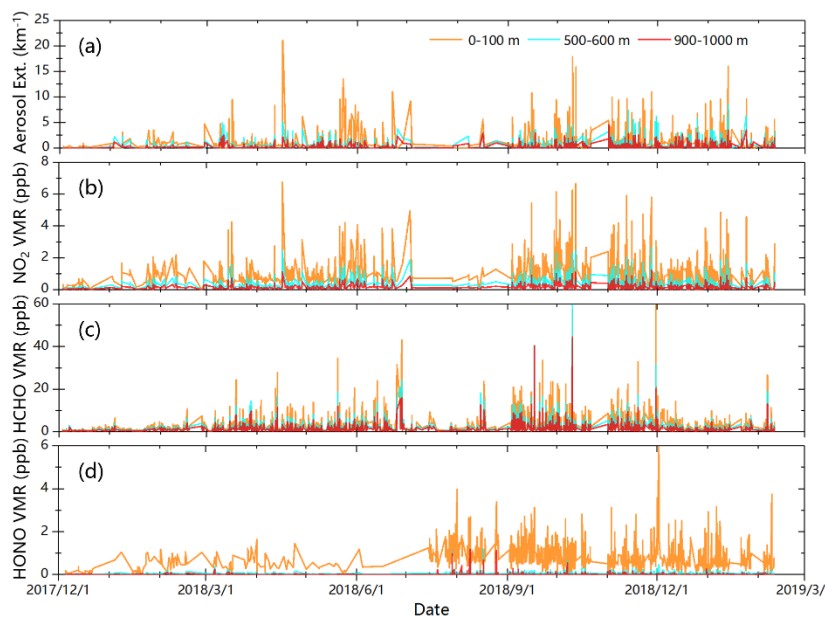

**Figure 6. Time series of hourly averaged (a) aerosol extinction, (b) NO2, (c) HCHO, and (d) HONO from December 2017 to February 2019 at the CAS (QOMS) station.**

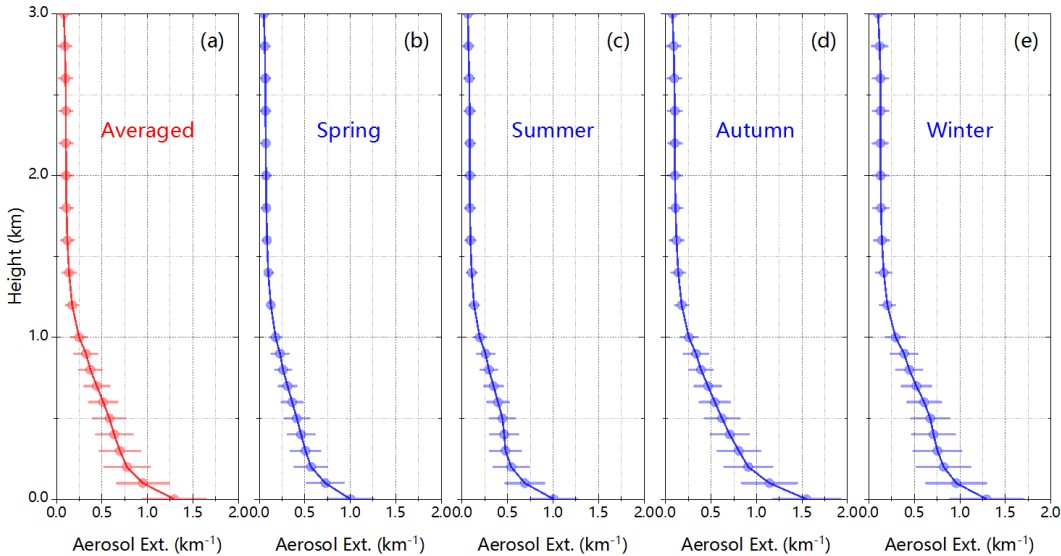

**Figure 7.** (a) total averaged, (b) spring averaged, (c) summer averaged, (d) autumn averaged, and (e) winter averaged aerosol
extinction vertical profiles from December 2017 to February 2019 at the CAS (QOMS) station.

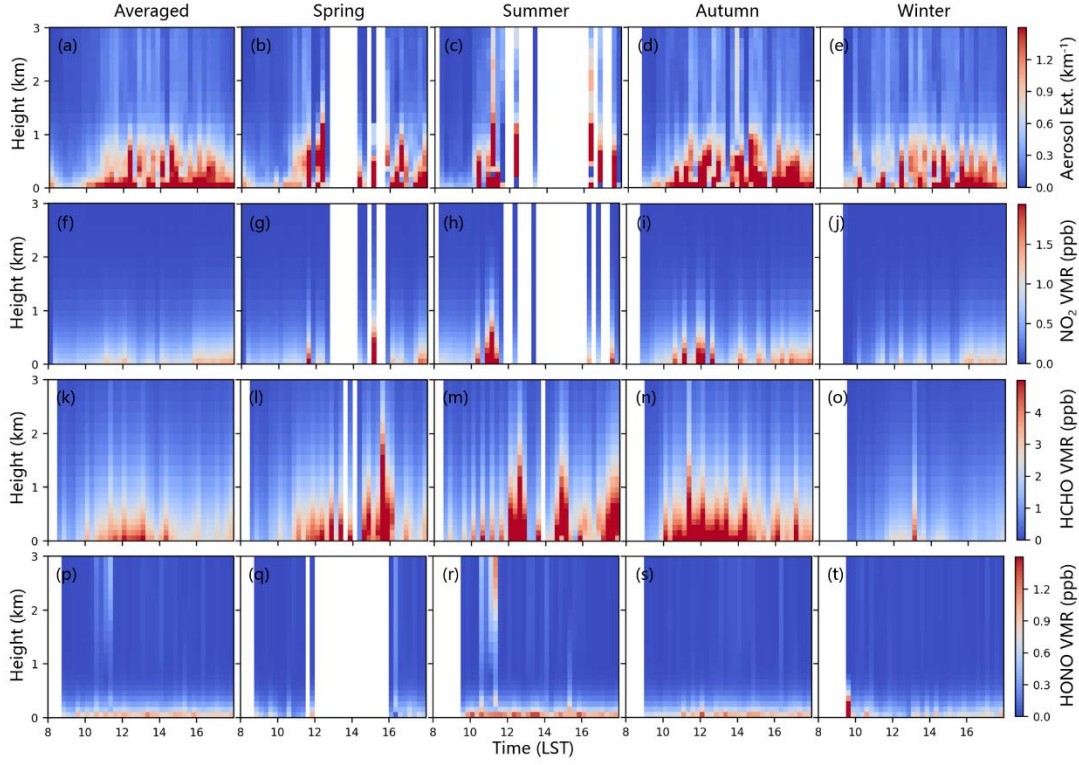

**Figure 8.** Diurnal variations of the total averaged and seasonal averaged aerosol extinction profiles (a-e), NO2 profiles (f-j), HCHO
profiles (k-o), and HONO profiles (p-t) from December 2017 to February 2019 at the CAS (QOMS) station.



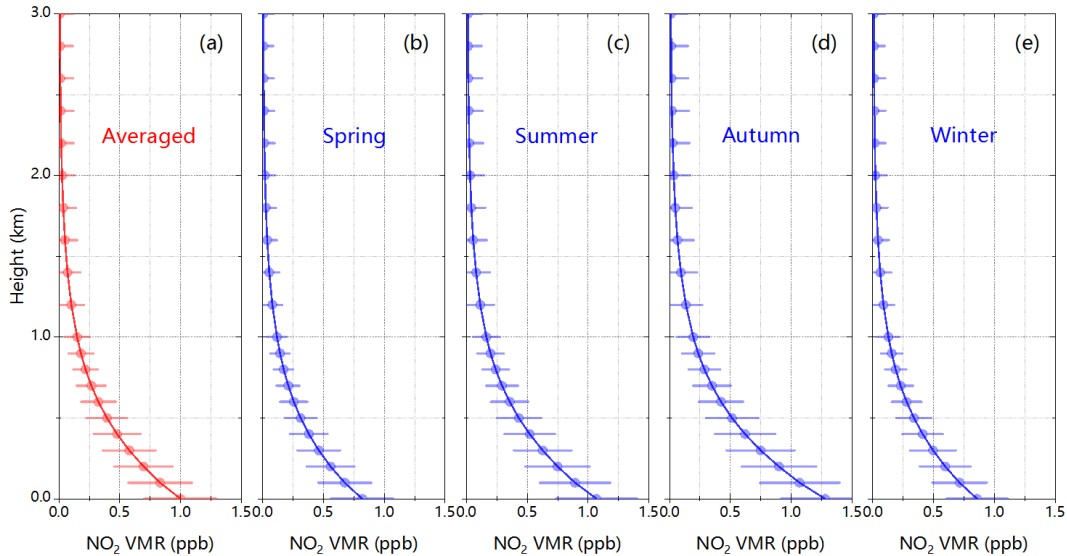

**Figure 9. (a) total averaged, (b) spring averaged, (c) summer averaged, (d) autumn averaged, and (e) winter averaged NO2 vertical profiles from December 2017 to February 2019 at the CAS (QOMS) station.**

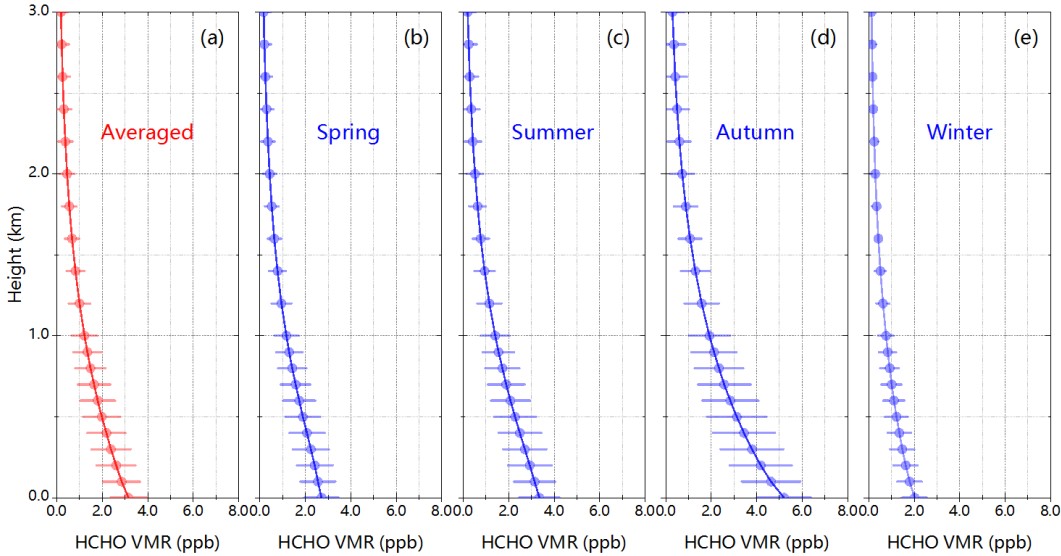

**Figure 10. (a) total averaged, (b) spring averaged, (c) summer averaged, (d) autumn averaged, and (e) winter averaged HCHO vertical profiles from December 2017 to February 2019 at the CAS (QOMS) station.**



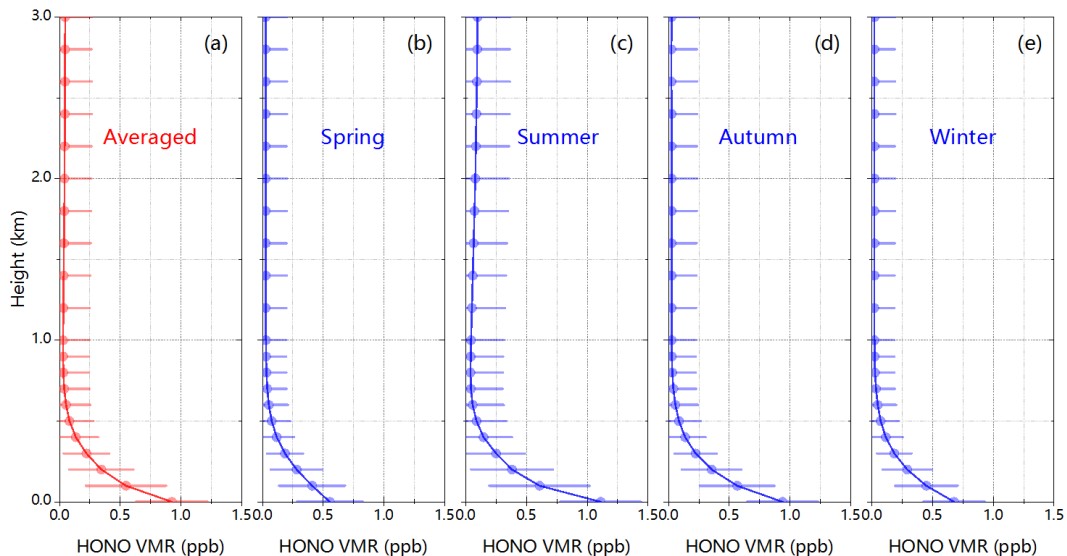

**Figure 11. (a) total averaged, (b) spring averaged, (c) summer averaged, (d) autumn averaged, and (e) winter averaged HONO vertical profiles from December 2017 to February 2019 at the CAS (QOMS) station.**

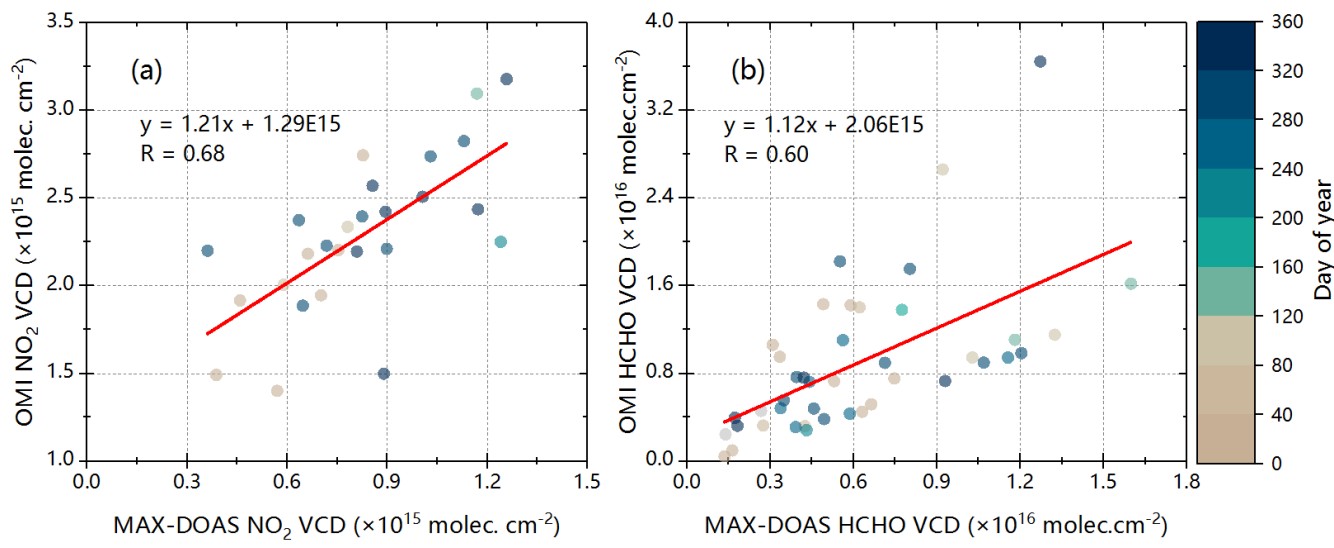

**Figure 12. Linear regression plots (a) of the tropospheric NO2 VCDs measured by OMI and MAX-DOAS, (b) of the tropospheric HCHO VCDs measured by OMI and MAX-DOAS.**