# Peer review of "Ground-based vertical profile observations of atmospheric composition on the Tibetan Plateau (2017-2019)"

_Earth System Science Data, 2021_

## Author Comment (AC1)

The authors described an innovative tool for ground-level vertical profile observations of atmospheric composition. The data could be used publicly. The manuscript is well-written.

Response: We would like to thank the reviewer for careful reading, and valuable suggestions. Point-to-point responses are given below. The original comments are in black and our responses are in blue.

Some minor comments as follows:

(1) It would also be great to know that limitation of this data so that the potential users could use the data appropriately.

Response: Thanks for your suggestion. We added limitation in the revised manuscript.

The limitations of this data are described as follows. (1) MAX-DOAS in ultraviolet and visible spectral ranges are typically affected by photon-shot noise, and the retrieval errors usually increase under heavy haze or cloudy conditions. The data with relative retrieval errors larger than 50% were filtered in this study; (2) only the daytime vertical profiles of aerosol, $NO_2$, HCHO and HONO were retrieved since MAX-DOAS relies on scattered sunlight. The spectral collected when solar zenith angle (SZA) are larger than $75^o$ were filtered to avoid the strong absorption of stratosphere (Xing et al., 2017); (3) the vertical resolution of 100 m is the highest resolution at present, which still needs to be improved with the development of hardware and algorithms in the future.

(2) The authors could also provide more information regarding how to potentially apply this dataset to other studies.

Response: Thanks for your suggestion. The potential applications of this dataset include: (1) Performances of chemical transport models are commonly poor when applied over the Tibet plateau due to complex topography and meteorology, etc. This dataset can be used to reduce the uncertainties of these models, especially in the vertical direction (Liu et al., 2021a); (2) this dataset can assist in the source apportionments of atmospheric composition at different altitudes over the Tibet Plateau; (3) observed atmospheric composition over the TP is valuable inputs for box models to understand atmospheric oxidation capacity at different altitudes on the Tibet Plateau.

(3) The methods for data retrieval were well written. However, it would also be great if there is a bit more details regarding why using these methods and how accurate these methods can be.

Re: Thanks for your suggestion. Three algorithms, namely parameterized, optimal estimation and look-up table, are currently widely used in this field. The optimal estimation algorithm often shows the highest sensitivity in vertical space (Frieß et al., 2017). Reasonably good agreements were previously found in validation of retrieved vertical profiles using optimal estimation method with lidar and balloon observations (Wang et al., 2019). We also provided here an example of retrieval errors, averaging kernel and degree of freedom of the retrieved profiles of aerosol, $NO_2$, HCHO and HONO in Figure R1, which has been inserted into the supplement.

[Figure]

Figure R1. Retrievals at 11:30 (LST) on 10 March at the CAS (QOMS) station. (a) aerosol extinction, (b) NO2, (c) HCHO and (d) HONO. The top row shows the retrieved profiles, plotted with their associated a priori profile and retrieved errors. The bottom row presents the averaging kernels and degrees of freedom for signal associated with the profile retrieval.

References

Xing, C., Liu, C., Wang, S., Chan, K. L., Gao, Y., Huang, X., Su, W., Zhang, C., Dong, Y., Fan, G., Zhang, T., Chen, Z., Hu, Q., Su, H., Xie, Z., and Liu, J.: Observations of the vertical distributions of summertime atmospheric pollutants and the corresponding ozone production in Shanghai, China, Atmos. Chem. Phys., 17, 14275–14289, https://doi.org/10.5194/acp-17-14275-2017, 2017.

Liu, C., Gao, M., Hu, Q., Brasseur, G.P., Carmichael, G.R.: Stereoscopic monitoring: a promising strategy to advance diagnostic and prediction of air pollution, Bull. Am. Meteoro. Soc., 102(4), E730-E737.

Frieß, U., Beirle, S., Bonilla, L.A., Bösch, T., Friedrich, M.M., Hendrick, F., Piters, A., Richter, A., van Roozendael, M., Rozanov, V.V., Spinei, E., Tirpitz, J.L., Vlemmix, T., Wagner, T., and Wang, Y.: Intercomparison of MAX-DOAS vertical profile retrieval algorithms: studies using synthetic data, Atmos. Meas. Tech., 12, 2155–2181, 2019.

Wang, Y., Dörner, S., Donner, S., Böhnke, S., De Smedt, I., Dickerson, R.R., Dong, Z., He, H., Li, Z., Li, Z., Li, D., Liu, D., Ren, X., Theys, N., Wang, Y., Wang, Y., Wang, Z., Xu, H., Xu, J., and Wagner, T.: Vertical profiles of $NO_2$, $SO_2$, HONO, HCHO, CHOCHO and aerosols derived from MAX-DOAS measurements at a rural site in the central western North China Plain and their relation to emission sources and effects of regional transport, Atmos. Chem. Phys., 19, 5417–5449, 2019.

---

## Author Comment (AC2)

This paper presents a unique dataset collected using the Multi-axis differential optical absorption spectroscopy (MAX-DOAS) at a site located on the Tibetan Plateau, where observations are extremely sparse. These vertical profile measurements of several key atmospheric compositions, i.e., AOD, $NO_2$, HCHO, and HONO, over a relatively long time period (Dec. 2017 ~ Mar. 2019), are very valuable to the scientific community and policy makers. Among many potential usages of the dataset, to constrain model representation and assist in satellite retrieval is the obvious imminent application. The paper is generally well-written and the collected data are readily accessible. This reviewer suggests acceptance for publication after the authors address the following minor comments.

Response: We would like to thank the reviewer for valuable inputs and careful reading. Point-to-point responses are offered below. The original comments are in black and our responses are in blue.

1. Under which conditions can the data be properly used?

Response:
We added the following information that describe the limitations of the data, and the data work under all other conditions:

(1) MAX-DOAS in ultraviolet and visible spectral ranges are typically affected by photon-shot noise, and the retrieval errors usually increase under heavy haze or cloudy conditions. The data with relative retrieval errors larger than 50% were filtered in this study;
We have filtered out the data observed under high cloud coverage according to the criterion of the color index (CI) being less than 10% of the threshold that obtained through fitting a fifth-order polynomial to CI data;
(2) only the daytime vertical profiles of aerosol, $NO_2$, HCHO and HONO were retrieved since MAX-DOAS relies on scattered sunlight. The spectral collected when solar zenith angle (SZA) are larger than 75° were filtered to avoid the strong absorption of stratosphere (Xing et al., 2017); (3) the vertical resolution of 100 m is the highest resolution at present, which still needs to be improved with the development of hardware and algorithms in the future.
(3) the data with relative retrieval errors larger than 50% were filtered;
We added this information in the summary section. Overall, we have filtered the data with greater uncertainties based on the above criterions. All data provided have been carefully checked, and they can be used with high confidence.

2. What is the estimated measurement uncertainty?

Response:
➢ In this dataset, we have calculated the retrieved errors to estimate the measurement uncertainties.
➢ "The profiles of aerosol and trace gases were filtered out when the degree of freedom (DFS) was less than 1.0 and retrieved relative error were larger than 100%."
➢ In the revised manuscript, we added an example of the retrieval errors, averaging kernel and degree of freedom of the retrieved profiles of aerosol, $NO_2$, HCHO and HONO in Figure S2.
➢ The uncertainty of NO2 is relatively larger than other three species, particularly near the ground.

[Figure]

> Figure S2. Retrievals at 11:11:30 (LST) on 10 March at the CAS (QOMS) station. (a) aerosol extinction, (b) NO2, (c) HCHO and (d) HONO. The top row shows the retrieved profiles, plotted with their associated a priori profile and retrieved errors. The bottom row presents the averaging kernels and degrees of freedom for signal associated with the profile retrieval.

3. The authors cross-check the data with OMI measured $NO_2$ and HCHO. Has any comparison with the satellite measured AOD been carried out? Since MAX-DOAS measures vertical profiles, the comparison with CALIPSO observed AOD profile will be informative and valuable.

Re: Thanks for your suggestion.
> MAX-DOAS only observed during the daytime, and CALIPSO overpass the area around CAS (QOMS) once every l6 days at about 07:00 (UTC). During the MAX-DOAS observation periods, a total of three CALIPSO orbits were discovered within 50 km from the CAS (QOMS). Unfortunately, the aerosol extinction coefficients of these three orbits near the CAS (QOMS) are invalid with the value of -999, which may be due to the large noise of the CALIPSO data during above three days. Above three CALIPSO orbits are:
- CAL_LID_L2_05kmAPro-Standard-V4-20.2017-12-13T07-01-59ZD_Subset.hdf
- CAL_LID_L2_05kmAPro-Standard-V4-20.2018-11-09T07-03-10ZD_Subset.hdf
- CAL_LID_L2_05kmAPro-Standard-V4-20.2019-01-05T07-01-13ZD_Subset.hdf
> In order to validate the MAX-DOAS aerosol vertical profile, we compared it against the aerosol vertical profiles monitored by Mie scattering lidar which also installed in CAS (QOMS). 08:30 02 August 2018 was selected as an example. In Figure S3, we could find these two aerosol profiles consistent, although MAX-DOAS observed extinction is slightly higher than that of Mie lidar.
> We also used Himawari-8 AOD to validate MAX-DOAS AODs. Due to the influence of high retrieval noise and high surface albedo in CAS (QOMS) and its around areas, the AOD observed by Himawari-8 is seriously missing from December 2017 to March 2019. Only 6 valid data were found, but we found that MAX-DOAS AOD and Himawari-8 AOD still show very good correlation (R=0.96).

[Figure]

Figure S3. Aerosol extinction vertical profiles measured by MAX-DOAS and Mie scattering lidar at 08:30 of 02 August, 2018.

[Figure]

Figure S4. Scatter plots of MAX-DOAS AOD against with Himawari-8 AOD.

4. Addition of some metadata or readme file on the data portal will better assist potential users in correctly interpreting and using this dataset.

Response: Thanks for your suggestion. We provide the vertical profiles of aerosol, $NO_2$, HCHO and HONO measured on the CAS (QOMS) from December 2017 to March 2019 in the dataset available at https://doi.org/10.5281/zenodo.5336460. The readme file have been added to this webpage.

(1) Aerosol:
● The fist column: Date_Time (UTC) with a format of YYYY/MM/DD_hh:mm;
● The 2-22 column: aerosol extinction coefficients with the unit of $km^{-1}$;
● The 23-43 column: errors of aerosol extinction coefficients with the unit of $km^{-1}$;
● The 44 and 45 columns are AODs and their corresponding errors;
● The unit of height: km.

(2) $NO_2$:
● The fist column: Date_Time (UTC) with a format of YYYY/MM/DD_hh:mm;
● The 2-22 column: the concentrations of $NO_2$ with the unit of ppb;

- The 23-43 column: errors of the retrieved concentration of $NO_2$ with the unit of ppb;
- The 44 and 45 columns are the tropospheric $NO_2$ VCDs and VCD errors with the unit of molec.cm$^{-2}$;
- The unit of height: km.

(3) HCHO:
- The fist column: Date_Time (UTC) with a format of YYYY/MM/DD_hh:mm;
- The 2-22 column: the concentrations of HCHO with the unit of ppb;
- The 23-43 column: errors of the retrieved concentration of HCHO with the unit of ppb;
- The 44 and 45 columns are the tropospheric HCHO VCDs and VCD errors with the unit of molec.cm$^{-2}$;
- The unit of height: km.

(4) HONO:
- The fist column: Date_Time (UTC) with a format of YYYY/MM/DD_hh:mm;
- The 2-22 column: the concentrations of HONO with the unit of ppb;
- The 23-43 column: errors of the retrieved concentration of HONO with the unit of ppb;
- The 44 and 45 columns are the tropospheric HONO VCDs and VCD errors with the unit of molec.cm$^{-2}$;
- The unit of height: km.

5. Line 270, change 'HCHO' to 'HONO'.

Response: Thanks for your suggestion. We have changed "HCHO" to "HONO" as you suggested.

6. Line 274, change 'Such as pattern' to 'Such pattern'.

Response: Thanks for your suggestion. We have changed "Such as pattern" to "This pattern" in the revised manuscript.

---

## Author Response (AR2)

Editor's comments:

please also cite the dataset in the introduction section like "Sect. 5 present the availability of this dataset (Xing et al., 2021) and ....."

Reply: We have added in the revised manuscript.